# Hierarchical Abstraction for Combinatorial Generalization in Object Rearrangement

Michael Chang,* Alyssa L. Dayan, Franziska Meier, Thomas L. Griffiths, Sergey Levine, Amy Zhang

## Abstract

Object rearrangement is a challenge for embodied agents because solving these tasks requires generalizing across a combinatorially large set of underlying entities that take the value of object states. Worse, these entities are often unknown and must be inferred from sensory percepts. We present a hierarchical abstraction approach to uncover these underlying entities and achieve combinatorial generalization from unstructured inputs. By constructing a factorized transition graph over clusters of object representations inferred from pixels, we show how to learn a correspondence between intervening on states of entities in the agent's model and acting on objects in the environment. We use this correspondence to develop a method for control that generalizes to different numbers and configurations of objects, which outperforms current offline deep RL methods when evaluated on a set of simulated rearrangement and stacking tasks.

## 1 Introduction

A core property of intelligence is the ability to re-purpose previously acquired knowledge for solving new problems, but how to identify symmetries for factorizing modeling and control into independent components has been a challenge for AI systems, which either assume human-defined abstractions to begin with [11, 13] or learn monolithic representations with no clear mechanism for reuse [19, 30].

The problem of *object rearrangement* offers an intuitive setting for studying this problem of knowledge reuse. Because the space of object configurations is combinatorially large, solving novel rearrangement problems requires the agent to recognize that the same action for moving an object from one location to another can be reused for different objects in different contexts. We can formulate this problem as an offline goal-conditioned reinforcement learning (RL) problem, where the agent is trained on a dataset of sensorimotor interactions and is evaluated on rearranging objects to satisfy constraints depicted in a goal image. The research questions are, therefore, how can we enable the agent to represent abstractions of objects in a way that is amenable for such reuse in control, and how can we enable it to discover these abstractions on its own directly from the sensorimotor interface?

Our main contribution is a method called Hierarchical Abstraction (HA) that offers an answer to the above questions. The key idea is to represent objects as independent and symmetric latents, which we call entities, and factorize each entity $h^k$ into two parts, its identity $z^k$ and its state $s^k$. The identity is the part that is not affected by action and the state is the part that is. The modeling component of our approach abstracts the dataset of sensorimotor interactions into a graph over state transitions of individual entities. By construction, a state transition is agnostic to the identity of the entity or the context entities present in the sensorimotor interaction from which it was learned, enabling it to be reused across multiple entities and contexts. The control component of our approach then re-composes sequences of state transitions to solve new rearrangement problems by inferring what state transition can be taken given only the current and goal image observations.

---

*work done as an intern at Meta AI. Correspondence to: mbchang@berkeley.edu and amyzhang@fb.com

4th Workshop on Shared Visual Representations in Human and Machine Visual Intelligence (SVRHM) at the Neural Information Processing Systems (NeurIPS) conference 2022. New Orleans.

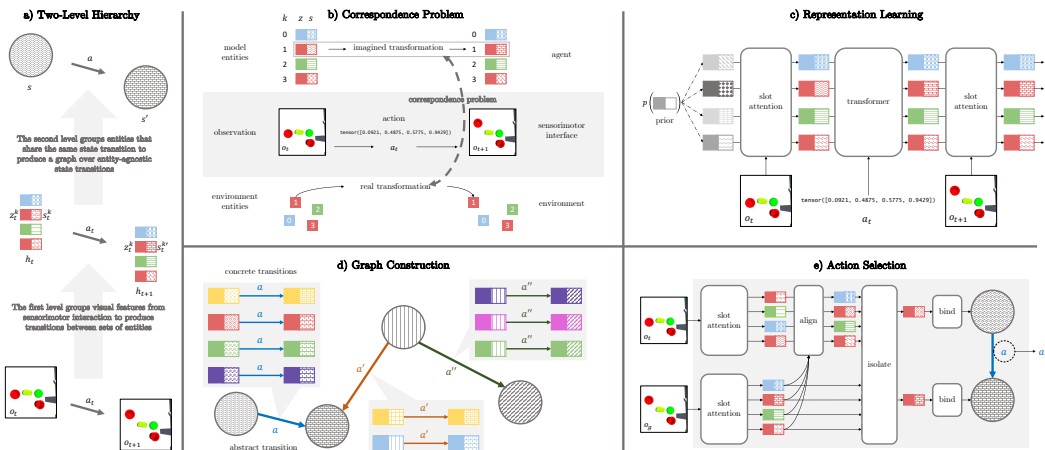

Figure 1: **Overview** (a) HA abstracts video interactions into a graph over state transitions of individual entities with a two-level hierarchy. (b) To do so requires learning representations of entities in a way that exhibits a correspondence between how model entities and how real objects transform under action. (c) The first level of the hierarchy abstracts visual features into transitions over sets of entities. (d) The second level abstracts these transitions into a graph of transitions over individual states. (e) HA decomposes the rearrangement task into a subtask per entity and chooses actions by returning the action tagged to the edge between nodes in the graph.

In contrast to other works in object-centric learning [14–16, 21, 24, 31, 34, 40] which evaluate the quality of inferred entities via segmentation metrics, we evaluate our inferred entities by *how well they can reused for solving tasks*, which more directly assesses what we want an agent's object representations to do. Kulkarni et al. [23], Veerapaneni et al. [36] also evaluate on control tasks. Whereas Kulkarni et al. [23] considers how object representations improve exploration, we consider the offline setting which requires zero-shot generalization. The shooting-based planning method in Veerapaneni et al. [36] suffers from compounding errors as other learned single-step models do [20], but our non-parametric approach of composing previously seen transitions enables us to plan for longer horizons. Indeed, our results show that HA outperforms both state-of-the-art offline RL methods and slot-based planning methods across three simulated object rearrangement problems.

## 2 Hierarchical Abstraction

We present our method for solving object rearrangement problems from pixels. Given a transition $o, a \rightarrow o'$ from the training set in which a single object $k$ has been moved, the modeling problem is to represent the state transition $s^k \rightarrow s^{k\prime}$ as decoupled from the identity $z^k$ of the affected entity and as decoupled from the other context entities $h^{\neq k}$ in the scene. The control problem is to compose previously seen state transitions to satisfy goal constraints on new object configurations. Though we do not assume the number of action primitives is finite, we assume the number of observed locations can be clustered into a set of finite groups.

### 2.1 Modeling

Our modeling approach abstracts the training dataset of action-conditioned videos into a factorized graph over state transitions that can be reused across different rearrangement problems. Constructing such a graph decomposes into two representation learning problems: the first concerns how to represent objects in a way that exposes their commonalities and the second concerns how to represent actions in a way that enables them to be reused across these commonalities. Our key insight is that these two problems are both different instances of the same fundamental problem of partitioning a space into discrete groups, but at different levels of abstraction: learning how to cluster visual features into entities (Fig. 1b) and learning how to cluster transitions over entity-sets into transitions over individual states (Fig. 1c). Our model thus implements a two-level abstraction hierarchy (Fig. 1a).

**Level 1: abstracting visual features into sets of entities** The goal of the first level is to map a transition over two video frames $o_1, a_1 \rightarrow o_2$ into a transition over entity-sets $(h_1^1, ..., h_1^K) \rightarrow (h_2^1, ..., h_2^K)$ (Fig. 1c). The criteria our entity-set transitions should satisfy are (1) entities represent

different objects in the scene, i.e. they are *independent* and *symmetric* and (2) only the state $s^k$ of entity $k$ changes, not its identity $z^k$, i.e. the entity is *factorized*. Because each video transition depicts only a single object being moved, we also want (3) the transition to be *sparse*, i.e. only one entity $h^k$ changes in the transition and the rest $h^{\neq k}$ remain unchanged.

Our approach manually enforces independence, symmetry, and factorization but lets sparsity emerge. Specifically, we formulate the inference of entity-sets as the application of an equivariant action-conditioned sequential Bayesian filter with a mixture model as the latent state, where entity representations are the parameters of the mixture components. Given the connection between the slots produced by slot attention (SA) [25] and mixture components Chang et al. [6], we concretely implement this filter by temporally evolving the slots produced by the slot attention module of the state-of-the-art SLATE architecture [32] with a transformer decoder (TD) [28, 35]. With $\mathbf{h} = (h^1, ..., h^K)$,

$$\mathbf{h}_{t+1} = SA(\hat{\mathbf{h}}_t, o_{t+1}) \qquad \hat{\mathbf{s}}_{t+1} = TD\left(\text{queries} = \mathbf{s}_t, \text{keys/values} = [\mathbf{s}_t, a_t]\right),$$

where $\hat{\mathbf{h}}_{t+1} = [\mathbf{z}_t, \hat{\mathbf{s}}_{t+1}]$ and $\hat{\mathbf{h}}_0$ are independent samples from a Gaussian. This approach enforces independence because $\hat{\mathbf{h}}_0$ are independent, symmetry because the transformer is equivariant, and factorization because $\hat{\mathbf{z}}_{t+1}$ is a copied from $\mathbf{z}_t$. Empirically, we find that this architecture models sparse changes in the slots (Fig. 3). We call this implementation **dynamic SLATE** (dSLATE), but our contribution of how structurally we enforce the above three criteria remains invariant to *how* the sequential Bayesian filter is implemented, which we expect will advance beyond SA and TD.

**Level 2: abstracting transitions over sets of entities into individual state transitions**   The goal of the second level is to construct a state transition graph from a buffer of entity-set transitions $\{(\mathbf{h}_1, ...\mathbf{h}_T)\}_{n=1}^N$ produced from the first level (Alg. 1, Fig. 1d). Specifically, we seek each edge of this graph to represent a state transition that has been shared across multiple entities in different contexts in the training dataset, such that we can reuse such state transitions when choosing actions for solving new rearrangement problems. Given our assumption that observed locations can be grouped into a finite set of clusters, this goal translates into a problem of clustering state transitions, and the problem of clustering state transitions reduces to clustering the states before and after the transition. We treat each cluster centroid as a node in the state transition graph, and an edge between nodes is tagged with the single action that transforms one node's state to another's.

To create the nodes, the first step is to identify the entity $h^k$ that dSLATE predicted was affected by $a_t$ in each transition $(\mathbf{h}_t, a_t, \mathbf{h}_{t+1})$. Our `isolate` procedure achieve this by solving $k = \arg\max_{k' \in \{1,...,K\}} d(s_t^{k'}, s_{t+1}^{k'})$ to identify the index of the entity whose state has most changed during the transition, where $d(\cdot, \cdot)$ is a distance function, detailed in the Appendix. Having the buffer of transitions over entity sets $\{(\mathbf{h}_1, ...\mathbf{h}_T)\}_{n=1}^N$ to a buffer of individual entity transitions $\{(h_t^k, a_t, h_{t+1}^k)\}_{n=1}^N$, and thus a buffer of individual state transitions $\{(s_t^k, a_t, s_{t+1}^k)\}_{n=1}^N$, the second step is to cluster similar transitions together by clustering the states

---

**Algorithm 1** Building the Graph

1: **input** `model`, `buffer`
2: **for** $\{(o_t, a_t, o_{t+1})\}_n$ in `buffer` **do**
3:     # infer entities from transition
4:     $\{(\mathbf{h}_t, a_t, \mathbf{h}_{t+1})\}_n \leftarrow$ `model` $(\{o_t, a_t, o_{t+1}\}_n)$.
5:     # identify which entity changed in transition
6:     $\{(h_t^k, a_t, h_{t+1}^k)\}_n \leftarrow$ `isolate` $(\{(\mathbf{h}_t, a_t, \mathbf{h}_{t+1})\}_n)$
7: **end for**
8: # partition transitions by clustering entities
9: $\{s_*\}_{m=1}^M \leftarrow$ `cluster` $\left(\{(s_t^k, a_t, s_{t+1}^k)\}_{n=1}^N\right)$
10: # transitions between clusters are edges
11: **initialize** `graph` with nodes $s_*^{[m]}$, for $m \in [1:M]$
12: **for each** $\{(h_t^k, a_t, h_{t+1}^k)\}_n$ **do**
13:     # infer cluster assignments
14:     $[i], [j] \leftarrow$ `bind` $(h_t^k)$, `bind` $(h_{t+1}^k)$
15:     # tag edge with action $a_t$
16:     `graph.edges`$[i, j] \leftarrow$ `create-edge` $\left(s_*^{[i]} \overset{a_t}{\to} s_*^{[j]}\right)$
17: **end for**
18: **return** `graph`

---

before and after each transition. Our `cluster` procedure achieves this by clustering over all states together, which is sufficient since the end state of transition is the starting state of another. A proper clustering should return centroids $\{s_*\}_{m=1}^M$ that all correspond to actual states of entities, which we hypothesize will correspond to actual locations in the training set based on how dSLATE is trained.

The purpose of edges between the nodes is to record an action that transforms an entity from one state to another. For each entity transition $(h_t^k, a_t, h_{t+1}^k)$ we `bind` $h_t^k$ and $h_{t+1}^k$ to their associated nodes $s_*^{[i]}$ and $s_*^{[j]}$ and create an edge between $s_*^{[i]}$ and $s_*^{[j]}$ tagged with action $a$, overwriting previous

edges based on the assumption that with a proper clustering there should only be one action primitive per pair of nodes. Applying the `bind` procedure to an entity $h^k$ returns the index of the cluster of the centroid $s_*$ that is nearest to the entity's state $s^k$. For our experiments `cluster` is implemented with K-means and `bind` is implemented with nearest neighbors using the same distance metric used for K-means (see Appendix), but other clustering algorithms and distance metrics can also be used.

## 2.2 Control

Having constructed a graph of state transitions, our approach for control re-composes sequences of state transitions to solve new rearrangement problems. Specifically, the agent decomposes the rearrangement problem into a set of per-entity subproblems, searches the transition graph for a transition that transforms the current entity's state to its goal state, and executes the action tagged with this transition in the environment. This problem decomposition is possible because the transitions in our graph are constructed to be agnostic to identity and context, enabling different rearrangement problems to share solutions to the same subproblems. The core challenge in deciding which transitions to compose is in determining which transitions are *possible* to compose. That is, the agent must determine which nodes in the graph correspond to the given goal constraints and which nodes correspond to the entities in the current observation, but the current entities $\mathbf{h}_t$ and goal constraints $\mathbf{h}_g$ must themselves be inferred from the current and goal observations $o_t$ and $o_g$, requiring the agent to infer both what to do and how to do it purely from its sensorimotor interface.

---

**Algorithm 2** Action Selection

1: **given** `model`, `graph`
2: **input** goal $o_g$, observation $o_t$
3:   # infer goal constraints and current entities
4:   $\mathbf{h_g}, \mathbf{h_t} \leftarrow \texttt{model}\left(o_g\right), \texttt{model}\left(o_t\right)$
5:   align entity indices of $\mathbf{h_t}$ with those of $\mathbf{h_g}$
6:   $\pi \leftarrow \texttt{align}\left(\mathbf{h_t}, \mathbf{h_g}\right)$
7:   permute indices of $\mathbf{h_t}$ according to $\pi$
8:   $\mathbf{h_t} \leftarrow \left(h_t^{\pi[1]}, ..., h_t^{\pi[K]}\right)$
9:   identify $k$th goal constraint to satisfy next
10:  $k \leftarrow \texttt{select-constraint}\left(\mathbf{h_t}, \mathbf{h_g}\right)$
11:  infer cluster assignments
12:  $[i], [j] \leftarrow \texttt{bind}\left(h_t^k\right), \texttt{bind}\left(h_g^k\right)$
13:  action that transforms node $[i]$ to node $[j]$
14:  **return** `graph.edges`$[i, j]$`.action`

---

Our approach takes four steps (Alg. 2, Fig. 3). The first step applies dSLATE to infer $\mathbf{h}_t$ and $\mathbf{h}_g$ from $o_t$ and $o_g$. The second step permutes the indices of $\mathbf{h}_t$ and $\mathbf{h}_g$ to `align` the identities of $h_t^k$ with $h_g^k$. This step is necessary because the permutation invariance of the entities does not guarantee that the entities with the same identities will have the same index $k$, i.e. $z_t^k$ might not correspond to $z_g^k$. We implement this using the Hungarian algorithm that minimizes the matching cost between $\mathbf{z}^t$ and $\mathbf{z}^g$. The third step selects which goal constraint $h_g^k$ to satisfy next. In the case where objects can move independently, we implement this `select-constraint` procedure by determining which constraint $h_g^k$ has the highest difference in state with its counterpart $h_t^k$, which reduces to solving the same argmax problem as in `isolate`. We discuss how we handle more complex dependencies in the Appendix. Lastly, we `bind` $h_t^k$ and $h_t^g$ to the graph and return the action tagged to the edge between their respective nodes. If an edge does not exist between the nodes, we simply take a random action.

## 3 Experiments

The crucial test for knowledge reuse is to solve a set of tasks from a part of the combinatorial space of object rearrangement problems that are disjoint from the part of the space given for training, i.e. generalizing to different numbers of objects. We measure the "fractional success rate:" the mean change in the number of satisfied constraints divided by the number of initially unsatisfied constraints for each episode. In the *complete* setting, all objects have an associated constraint, whereas in the *partial* setting, only a subset of objects have associated constraints, and this is reflected in the goal image (Fig. 2b). We consider three challenging environments: *block-rearrange*, *robogym-rearrange*, and *block-stacking*. We compare against five baselines. We implement a version of MPC [36] by replacing our factorized graph search with CEM. We investigate an ablation of HA that constructs a monolithic graph over transitions between sets of slot states rather than individual slot states (MGS). We also compare with state-of-the-art pixel-based behavior cloning (BC) and implicit Q-learning (IQL) implementations based off of [22]. Our last baseline just takes random actions (Rand).

**Results** Figure 4 shows that HA performs significantly better than the baselines in combinatorial generalization (about a 5-10x improvement). Notably, the MPC baseline performs poorly because its rollouts are poor, and it is significantly more computationally expensive to run (11 hours instead of

20 minutes). Fig. 3a shows a t-SNE plot of the factors over 100 batches in the training set, where different colors label different clusters, with several clusters annotated with the average slot attention mask for the entities in the cluster. The right panel of Fig. 3 walks through the steps our algorithm takes to select an action, visualizing the `align`, `isolate`, and `bind` procedures, showing that the entities inferred with dSLATE intuitively capture objects that can be independently acted upon.

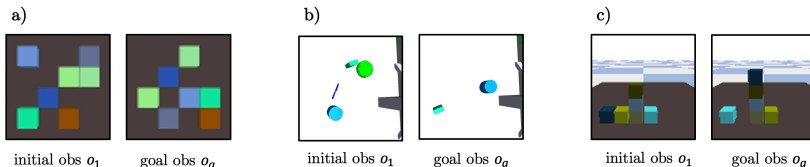

Figure 2: **Environments** Our environments are (a) *block-rearrange*, (b) *robogym-rearrange*, and (c) *block-stacking*. (a) shows the complete specification of goal constraints for all objects; (b) shows a partial specification of goal constraints. (c) shows the case where some goal constraints are already satisfied in the initial observation.

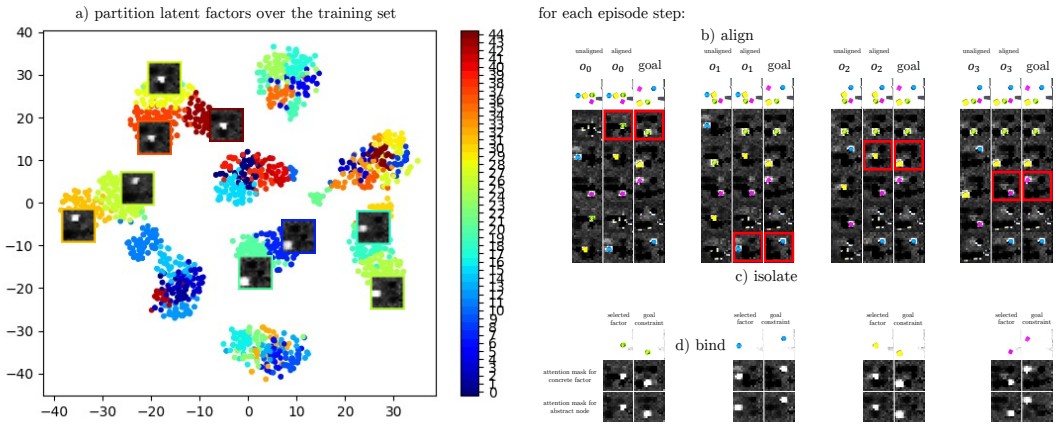

Figure 3: **Qualitative walk-through.** (a) shows the slot attention masks of the centroids from clustering states in *robogym-rearrange* (b) shows a walk-through of how HA selects the next action.

Figure 4: Generalizing from an offline dataset of 4 objects to solving rearrangement tasks with 4-7 objects, evaluated for 100 episodes across 10 seeds, showing standard error.

Table 1: *block-rearrange*, complete setting.

| Method | 4 | 5 | 6 | 7 |
|--------|---|---|---|---|
| Ours | **0.94** $\pm$ 0.01 | **0.93** $\pm$ 0.00 | **0.93** $\pm$ 0.00 | **0.89** $\pm$ 0.00 |
| Rand | 0.06 $\pm$ 0.02 | 0.07 $\pm$ 0.03 | 0.07 $\pm$ 0.03 | 0.08 $\pm$ 0.03 |
| MPC | 0.16 $\pm$ 0.06 | 0.12 $\pm$ 0.04 | 0.11 $\pm$ 0.04 | 0.10 $\pm$ 0.03 |
| MGS | 0.07 $\pm$ 0.03 | 0.06 $\pm$ 0.02 | 0.07 $\pm$ 0.02 | 0.08 $\pm$ 0.03 |
| IQL | 0.07 $\pm$ 0.01 | 0.03 $\pm$ 0.00 | 0.02 $\pm$ 0.00 | 0.02 $\pm$ 0.00 |
| BC | 0.03 $\pm$ 0.00 | 0.02 $\pm$ 0.00 | 0.01 $\pm$ 0.00 | 0.01 $\pm$ 0.00 |

Table 2: *block-stacking*, complete setting.

| Method | 4 | 5 | 6 | 7 |
|--------|---|---|---|---|
| Ours | **0.58** $\pm$ 0.02 | **0.48** $\pm$ 0.02 | **0.35** $\pm$ 0.01 | **0.24** $\pm$ 0.01 |
| Rand | 0.02 $\pm$ 0.01 | 0.01 $\pm$ 0.01 | 0.02 $\pm$ 0.01 | 0.02 $\pm$ 0.01 |
| MPC | 0.03 $\pm$ 0.01 | 0.01 $\pm$ 0.01 | 0.02 $\pm$ 0.01 | 0.01 $\pm$ 0.01 |
| MGS | 0.13 $\pm$ 0.00 | 0.01 $\pm$ 0.01 | 0.02 $\pm$ 0.01 | 0.01 $\pm$ 0.01 |
| IQL | 0.19 $\pm$0.02 | 0.12 $\pm$0.01 | 0.13 $\pm$0.02 | 0.07 $\pm$0.01 |
| BC | 0.21 $\pm$0.01 | 0.13 $\pm$0.01 | 0.13 $\pm$0.01 | 0.08 $\pm$0.01 |

Table 3: *robogym-rearrange*, complete setting.

| Method | 4 | 5 | 6 | 7 |
|--------|---|---|---|---|
| Ours | **0.64** $\pm$ 0.01 | **0.47** $\pm$ 0.01 | **0.49** $\pm$ 0.01 | **0.41** $\pm$ 0.01 |
| Rand | 0.01 $\pm$ 0.0 | 0.01 $\pm$ 0.0 | 0.0 $\pm$ 0.0 | 0.0 $\pm$ 0.0 |
| MPC | 0.0 $\pm$ 0.0 | 0.0 $\pm$ 0.0 | 0.0 $\pm$ 0.0 | 0.0 $\pm$ 0.0 |
| MGS | 0.01 $\pm$ 0.0 | 0.01 $\pm$ 0.0 | 0.0 $\pm$ 0.0 | 0.0 $\pm$ 0.0 |
| IQL | 0.0 $\pm$ 0.0 | 0.0 $\pm$ 0.0 | 0.0 $\pm$ 0.0 | 0.0 $\pm$ 0.0 |
| BC | 0.0 $\pm$ 0.0 | 0.0 $\pm$ 0.0 | 0.0 $\pm$ 0.0 | 0.0 $\pm$ 0.0 |

Table 4: *block-stacking*, partial setting.

| Method | 4 | 5 | 6 | 7 |
|--------|---|---|---|---|
| Ours | **0.34** $\pm$ 0.01 | **0.27** $\pm$ 0.01 | **0.21** $\pm$ 0.01 | **0.15** $\pm$ 0.01 |
| Rand | 0.03 $\pm$ 0.01 | 0.03 $\pm$ 0.01 | 0.03 $\pm$ 0.01 | 0.03 $\pm$ 0.01 |
| MPC | 0.04 $\pm$ 0.02 | 0.03 $\pm$ 0.01 | 0.02 $\pm$ 0.01 | 0.04 $\pm$ 0.01 |
| MGS | 0.03 $\pm$ 0.01 | 0.03 $\pm$ 0.01 | 0.02 $\pm$ 0.01 | 0.02 $\pm$ 0.01 |
| IQL | 0.14 $\pm$0.01 | 0.06 $\pm$0.01 | 0.04 $\pm$0.00 | 0.03 $\pm$0.01 |
| BC | 0.17 $\pm$0.01 | 0.07 $\pm$0.01 | 0.04 $\pm$0.00 | 0.03 $\pm$0.00 |

## 4 Discussion

We have demonstrated that HA outperforms various offline RL methods across three simulated object rearrangement problems. We hope HA will inspire future work on how abstractions of independent and symmetric entities can be learned and re-composed.

## Acknowledgments

We would like to thank Leslie Kaelbling for valuable feedback and Yash Sharma and Yilun Du for valuable discussions. This material is supported in part by the Fannie and John Hertz Foundation, as well as with ONR grant #N00014-18-1-2873.

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

# A   Additional Related Work

**Structured MDPs.** Many works have postulated that designing algorithms for structured MDPs will lead to improvements in sample efficiency and generalization over existing algorithms for the standard MDP formulation. The two closest types of MDP families to our work are factored MDPs [3, 4, 17], relational MDPs [37], and object-oriented MDPs [9]. Factored MDPs [3, 4, 17] assume that the environment can be represented by discrete attributes, and that transitions between these attributes can be modeled as a Bayesian network. Our work differs from these in that we do not assume access to these attributes and the dependency graph is also not assumed to be known. More importantly, the focus in this work is on using attributes to factorize a task into subcomponents; and in particular being able to generalize to new, more complex tasks at test time. Our approach is also related to Relational MDPs and Object-Oriented MDPs [1, 9, 12], where states are described as a set of objects, each of which is an instantiation of canonical classes, and each instantiated object has a set of attributes. Our work is especially related to [18], where the aim is to show that by using a relational representation of an MDP, a policy from one domain can generalize to a new domain through planning. However, we discover both objects (identities) and attributes (states), and achieve generalization through factorized planning, which grows linearly (rather than polynomially) in the number of factors.

# B   Implementation Details

This section details the implementation design decisions for each component of HA.

## B.1   Level 1: abstracting visual features into sets of entities

Dynamic SLATE (dSLATE) maps a video demonstration to a sequence of transitions over sets of entities. It consists of two main components: SLATE [32] and a transformer [28, 35] dynamics model. Because SLATE itself also uses a transformer for generating image reconstructions, dSLATE uses two transformers: one to recover the observation model $E$ and one to recover the dynamics model $P$ of the structured MDP. We use SLATE $\prod_k \mathcal{H}^k \times \mathcal{O} \to \prod_k \mathcal{H}^k$ to infer entities $\mathbf{h}_t$ from observation $o_t$ and initial guess $\hat{\mathbf{h}}_t$. The entity $h^k$ is also referred to as a *slot* in [25, 32] and is split in half as $h^k = (z^k, s^k)$. The first guess for each entity $\hat{h}_1^k$ is sampled independently and identically distributed from a unit Gaussian, whose parameters are also trained. The hyperparameters are given in Tab. 5.

SLATE preprocesses the image with a discrete variational autoencoder [29] into a grid of image features, encodes these features into a grid of tokens, infers slots from this token grid with Slot Attention [25], which also produces an attention mask `attn` over the features each slot attends to. These slots are trained using a transformer decoder [28, 35] to autoregressively reconstruct the tokens using the slots as keys/values.

We use the dynamics model $\prod_k \mathcal{S}^k \times \mathcal{A} \to \prod_k \mathcal{S}^k$ to predict a guess for $\hat{\mathbf{s}}_{t+1}$ given the action $a_t$ and the inferred entities $\mathbf{s}_t$ from the previous time-step. This dynamics model is implemented also as a transformer decoder, taking the entity states as queries, and the entity states and action as keys/values. This enables us the dynamics model to model the future state of the slots as an equivariant function of how the action affects it and of how it interacts with other entity states.

We trained dSLATE on an offline dataset of 5000 video demonstrations of length five, with each frame transition showing one of four objects being moved to a different location. We used five slots, one more than the number of objects, following the convention used in Van Steenkiste et al. [34], Veerapaneni et al. [36].

## B.2   Level 2: abstracting transitions over sets of entities into individual state transitions

HA constructs a graph of state transitions from a buffer of transitions over entity sets produced by dSLATE, as described in Alg. 1. Each transition records the identity $z$, state $s$, and `attn` of each entity of the entity sets $\mathbf{h}_t$ and $\mathbf{h}_{t+1}$. We also record the *pre-condition* of the transition, which we explain further in Appdx. B.3. Both $s$ or `attn` can serve as the representation of the state. We found that we obtained better clusterings when we used `attn` as the state for the *block-\** tasks and $s$ as the state for the *robogym-rearrange* task. We also empirically found that certain choices of distance metric used for K-means `clustering` and `binding` (implemented as nearest-neighbors) depended

| | | |
|---|---|---:|
| Number of epochs | | 200 |
| Episodes per epoch | | 5K |
| Episode length | | 5 |
| Batch size | | 32 |
| Peak LR | | (see caption) |
| LR warmup steps | | 30000 |
| Dropout | | 0.1 |
| Discrete VAE | Vocabulary Size | 4096 |
| | Temp. Cooldown | 1.0 to 0.1 |
| | Temp. Cooldown Steps | 30000 |
| | LR (no warmup) | 0.0003 |
| | Image Size | (see caption) |
| | Image Tokens | Image Size / 4 |
| transformer decoder | Layers | 4 |
| | Heads | 4 |
| | Hidden Dim. | 192 |
| Slot attention | Slots | 5 |
| | Iterations | 3 |
| | Slot Heads | 1 |
| | Slot Dim. ($h^k$) | 192 |
| | Identity Dim. ($z^k$) | 96 |
| | State Dim. ($s^k$) | 96 |
| transformer dynamics | Layers | 4 |
| | Heads | 4 |
| | Hidden Dim. | 96 |

Table 5: **Hyperparameters for training dSLATE** These hyperparameters are almost identical to those found in Singh et al. [32, Fig. 7], but because dSLATE operates on video demonstrations rather than static images, we changed some hyperparameters to save memory cost. We changed the batch size from 50 to 32, the number of transformer layers and heads from 8 to 4, the number of slot attention iterations from 7 to 3 without observing a significant change in performance. We used a peak learning rate of 0.0002 and an image size of 64 for *\*-rearrange*. We used a peak learning rate of 0.0003 and an image size of 96 for *block-stacking*.

on which choice of state representation we used, and this is summarized in Table 6. The K-means implmentation is adapted from `https://github.com/overshiki/kmeans_pytorch`. We found that increasing the number of slot attention iterations improved the entities representations especially when generalizing to more numbers of objects, so even though we dSLATE trained with slot attention three iterations, for inferring the slots from the buffer we used seven iterations. Lastly, we found that the number of clusters used to for K-Means is the most important hyperparameter for creating a graph that reflected the state transitions. We swept over 16 to 50 clusters and report the optimal number of clusters we found in Table 7.

| State representation | `attn` | $s$ |
|---|---|---|
| `isolate` distance metric $d(\cdot, \cdot)$ | cosine | cosine |
| `cluster` distance metric | IoU | squared Euclidean |
| `bind` distance metric | cosine | squared Euclidean |

Table 6: **Hyperparameters for constructing the transition graph with HA**

| | *block-rearrange* | *robogym-rearrange* | *block-stacking* |
|---|---|---|---|
| number of clusters | 30 | 45 | 47 |

Table 7: **Number of clusters used for constructing the nodes of the transition graph.**

## B.3  Action selection

Using dSLATE and the transition graph from HA, HA returns which action to execute in the environment given a goal and current observation, as described in Alg. 2. It first infers goal constraints $\mathbf{h}_g$ and current entities $\mathbf{h}_t$ from the goal observation $o_g$ and current observation $o_t$. It then uses the `align` procedure to align the indices of the entities in $\mathbf{h}_g$ and $\mathbf{h}_t$ and uses the `select-constraint` to choose the index $k$ of the entity to affect. It binds $h_t^k$ and $h_g^k$ to the graph and returns the action associated with the edge between their respective nodes. Because HA is a non-parameteric method, it could be the case the graph does not contain such an edge. In this case, we sample a random action in the environment, but future work will replace this step with a more sophisticated method.

To implement `align` we use the `scipy.optimize.linear_sum_assignment` implementation of the Hungarian algorithm, with Euclidean distances between the $z^k$'s as the matching cost.

In `select-constraint`, we are given the set of current entities $\mathbf{h}_t$ whose indices are aligned with the goal constraints $\mathbf{h}_g$ and returns the index $k$ of the goal constraint to satisfy next. By HA' construction, the edge between the nodes that $h_t^k$ and $h_g^k$ are bound to is the state transition that would be executed if the action associated to the edge were taken in the environment. The `select-constraint` procedure consists of three steps: (1) filtering possible transitions from impossible transitions (2) ranking transitions (3) sampling a transition.

**Filtering**    The filtering step implements HA' model of possibility and impossibility. In the filtering step, we consider, for each $k$, the transition between the nodes that $h_t^k$ and $h_g^k$ are bound to and mark the transition as possible or impossible. It then returns the indices $k$ over $\mathbf{h}_t$ and $\mathbf{h}_g$ whose associated transition from $h_t^k$ and $h_g^k$ is possible.

According to HA, a state transition between node $[i]$ and node $[j]$ is *possible* if its preconditions are met and there exists an edge for that state transition in the graph, and *impossible* otherwise. An intuitive example of an impossible transition is to place a block midair at some intended height, but this transition becomes possible if prior to the transition there already exists a stack of blocks that would support the block if the block were to be placed at that intended height. The existence of this supporting stack is thus the *precondition* for the transition to occur, rendering the stacked block *dependent* on the blocks supporting it.

When there are no dependencies among the entities, as in the *\*-rearrange* tasks where any object can be moved to any open location without considering where other objects are, any transition present in the graph is possible. When there are dependencies among the entities, as in *block-stacking*, we take the precondition of the transition into account. Although a precondition of a transition from node $[i]$ to node $[j]$ could be a function of both the source node $[i]$ and destination node $[j]$, for simplicity in this paper we consider preconditions as only a function of the destination node $[j]$, which rules out the possibility of placing a block in midair like the above example.

Because a precondition in general is a set of constraints that need to be satisfied for the transition to be possible, we represent the precondition of a transition into node $[j]$ (denoting the state $s_*^{[j]}$) as the set of context states $s_*^{[j']}$ that are always present whenever the state $s_*^{[j]}$ is present. Concretely, a block at height 3 at location $x$ is always accompanied by the presence of some block at height 2 and some block at height 1, both at location $x$; the states denoting (location $x$, height 2) and (location $x$, height 1) are the context states of the state (location $x$, height 3) and therefore serves as its precondition. We thus implement the precondition of a transition into node $[j]$ by recording the indices $j'$ of the nodes of states that are always present when node $[j]$ is a destination node. To test whether a precondition is satisfied for a given scene, we check if all nodes in the precondition set have a corresponding concrete entity that can be bound to it.

**Ranking**    The filtering step removes the indices from the entities $\mathbf{h}_t$ and goal constraints $\mathbf{h}_g$ whose transitions are impossible, yielding a possibly smaller set of entities $\tilde{\mathbf{h}}_t$ and constraints $\tilde{\mathbf{h}}_g$. That is, if $|\mathbf{h}_t| = |\mathbf{h}_g| = K$, then $|\tilde{\mathbf{h}}_t| = |\tilde{\mathbf{h}}_g| = \tilde{K} \leq K$.

The goal of the ranking step is to compute a ranking among the indices of $\tilde{\mathbf{h}}_t$ and $\tilde{\mathbf{h}}_g$ for choosing which index $k$ to actually select to affect with an action. Intuitively, we should rank indices $k$

according to how different $s_t^k$ and $s_g^k$ are because a difference indicates that the constraint $h_g^k$ is not satisfied. We reuse the distance metric $d(\cdot, \cdot)$ used for `isolate` to implement this ranking.

**Sampling** The goal of the sampling step is to select a $k \in \{1, ..., \tilde{K}\}$ whose associated entity we will affect with an action, given our ranking. One way to do this is to simply choose $k$ as $k = \arg\max_{k' \in \{1,...,\tilde{K}\}} d(s_t^{k'}, s_{t+1}^{k'})$ as in `isolate`, but we empirically found that sampling $k$ from a Categorical distribution whose pre-normalized probabilities are given by $d(s_t^{k'}, s_{t+1}^{k'})$ resulted in better task performance so we used this stochastic sampling approach. One explanation for why using the argmax may be worse is that it relies on the distance metric $d(\cdot, \cdot)$, and the state representation $s$, to be such that the distance metric flawlessly assigns high value to entities $k$ that need to be moved and low value to entities $k$ that do not need to be moved. But because the state space $\mathcal{S}$ is learned through the dSLATE training process without explicit supervision on the geometry of the space, a pair of points that should be farther apart than another set of points may not be accurately reflected by using a fixed distance metric $d(\cdot, \cdot)$. Future work will investigate imposing explicit supervision on the geometry of $\mathcal{S}$.

## C  Baseline Implementation Details

**Random (Rand)** The random policy takes actions using `env.action_space.sample()`.

**Behavior cloning (BC)** This approach trains a policy to output the actions directly taken in the provided dataset. We use an MSE loss to train the policy to imitate the actions.

**Implicit Q-learning (IQL)** IQL is a simple, offline RL approach that uses temporal difference (TD) learning with the dataset actions and trains a behavior policy value function. To produce an optimal value function, IQL estimates the maximum of the Q-function using expectile regression with an asymmetric MSE using the following objectives:

$$L_V(\psi) = \mathbb{E}_{(s,a) \sim \mathcal{D}}[L_2^\tau(Q_{\hat\theta}(s,a) - V_\psi(s))] \text{ where } L_2^\tau(u) = |\tau - \mathbb{1}(u < 0)|u^2 \tag{1}$$

$$L_Q(\theta) = \mathbb{E}_{(s,a,s') \sim \mathcal{D}}[(r(s,a) + \gamma V_\psi(s') - Q_\theta(s,a))^2] \tag{2}$$

$$L_\pi(\phi) = \mathbb{E}_{(s,a) \sim \mathcal{D}}[\exp(\beta(Q_{\hat\theta}(s,a) - V_\psi(s))) \log \pi_\phi(a|s)]. \tag{3}$$

The $V(s)$ estimates are used for TD-backups and the optimal policy is extracted with advantage-weighted behavioral cloning.

**Model predictive control (MPC)** This approach uses model predictive control with the cross entropy method (CEM) to select actions, using the transformer dynamics model of dSLATE to perform rollouts in latent space. This is similar to the approached used in OP3 [36], except that we use more recently proposed architectural components (slot attention [25] instead of IODINE [15], a transformer instead of a graph network [2, 7, 34]) so our MPC results are not directly comparable to that of OP3. We use the same dSLATE checkpoint that was used for HA.

We implement this MPC baseline using the `mbrl-lib` library [27] with 10 CEM iterations, an elite ratio of 0.05, and a population size of 250 which was the best configuration we found that fit within a wall clock budget of two days for 8 objects and 100 test episodes. We swept over CEM iterations of $[5, 10, 20]$, elite ratio of $[0.05, 0.1, 0.2]$, and population sizes of $[250, 500, 1000]$, and found that the elite ratio was the most important hyperparameter.

The cost function is computed by first aligning the predicted slots $\mathbf{h}_T$ and goal constraints $\mathbf{h}_g$ using the same `align` procedure in B.3, and then adding up the squared Euclidean distance between slots as $cost = \sum_k (h_T^k - h_g^k)^2$.

**Monolithic graph search (MGS)** This approach is an ablation to HA that does not construct a graph over state transitions of individual entities but instead constructs a graph over state transition over entity sets, i.e. each transition is $(\mathbf{s}, a, \mathbf{s}')$ rather than $(s^k, a, s^{k'})$. As with MPC, we use the same dSLATE checkpoint that was used for HA.

The purpose of this ablation is to elucidate the benefit of factorizing the transition graph over entities rather than entity sets. Because nodes in this transition graph represent a set of entity states rather

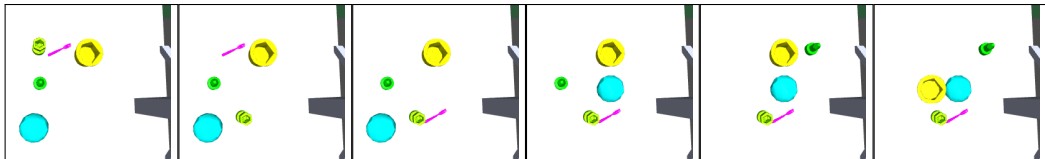

Figure 5: An example of solving a task in the robogym rearrange environment used in this paper.

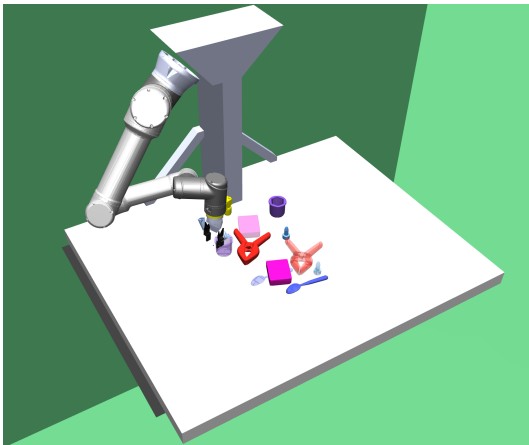

Figure 6: The original Robogym rearrange setup

than individual entity states, we use Dijkstra's algorithm, as in [10, 38, 39] to plan a unbroken path from the node the initial observation is bound to to the node a goal observation is bound to. For each time-step, we plan a path along the nodes using Dijkstra's algorithm, then return the action associated with the first edge along that path. Like HA, MGS is a non-parametric model, which means that for a set of entities to be bound to a node in the graph, that node must contain the exact set of entity states corresponding to the states of the entities. If we do not successfully bind to the graph, or if we do not find a path between the current node and the goal node, we sample a random action as HA does.

## D    Environment Details

*Block-rearrange* is our simplest environment, and *robogym-rearrange* and *block-stack* each add complexity along different axes. States and identities correspond to object locations and appearance respectively. In *block-rearrange* (Fig. 2a), all objects are the same size, shape, and orientation. $\mathcal{S}$ covers 16 locations in a grid. $\mathcal{Z}$ is the continuous space of red-green-blue values from 0 to 1. *robogym-rearrange* (fig. 2b) adapts the rearrange environment from OpenAI [26] and removes the assumption from *block-rearrange* that all objects are the same size, shape, orientation, and the assumption of predefined locations. *block-stack* (fig. 2c) adds preconditions and postconditions on whether objects can be moved: blocks can only be picked if from the top of a stack, and blocks can only be placed at a given height if there is an object beneath to support it (otherwise it falls). *Block-rearrange* and *Block-stack* are implemented in PyBullet [8] while *Robogym-rearrange* is implemented in Mujoco [33].

**Environments**    In *block-rearrange*, all objects are the same size, shape, and orientation and can exist in any one of 16 locations in a grid. Colors are sampled in a continuous space of red-green-blue values in $[0, 1]$.

The *robogym-rearrange* environment (see figures 5 and 6) is adapted from the rearrange environment in OpenAI's Robogym simulation framework [26] and removes the assumption from *block-rearrange* that all objects are the same size, shape, and orientation and the assumption of predefined locations. Furthermore, due to 3D perspective, the objects can look slightly different in different locations. The objects are uniformly sampled from a set of 94 meshes consisting of the YCB object set [5] and a set of basic geometric shapes, with colors sampled from a set of 13. The camera angle is a bird's eye view over the table, and the size of each object is normalized by its longest dimension, so tall

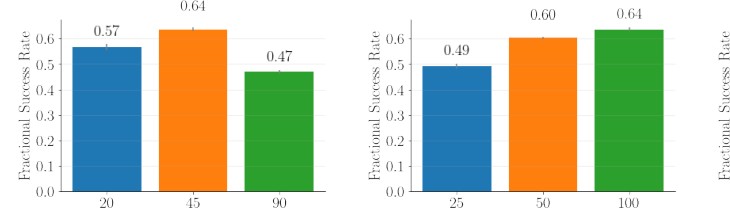

Figure 7: The performance of our method as the number of initialized clusters and batches from the training set used to construct the graph, and the number of slots are varied.

thin objects appear smaller. The objects' target positions are randomly sampled such that they don't overlap with each other or any of the initial positions, and the target orientation is set to be unchanged. Due to the continuous nature of this environment, we define a match threshold of at most 0.05 for both the initial pick position and the goal placement (the table dimensions are 0.6 by 0.8).

The *block-stack* environment adds preconditions and postconditions on whether objects can be moved: blocks can only be picked if they are at the top of a stack, and blocks can only be placed at a given height if there is already an object beneath to support it (otherwise it falls).

**Sensorimotor interface**  Each observation is a tuple of an initial image displaying the current observation and a goal image displaying constraints to be satisfied – the goal locations of the objects. Each action is a tuple $(w, \Delta w)$, where $w$ is a three-dimensional Cartesian coordinate $(x, y, z)$ in the environment arena.

For the $*$-*rearrange* tasks, objects are initialized at random non-overlapping locations that also do not overlap with their goal locations. For these tasks the $z$ (height) coordinate is always fixed. For the *block-stack* task, the goal locations are generated by randomly picking objects from the tops of stacks and placing them on other stacks. For this task the $y$ (depth) coordinate is always fixed.

An object is picked if $w$ is within a certain threshold of its location. For *block-$*$* where object locations are fixed points in a grid the object is snapped to the nearest grid location to $w + \Delta w$. Constraints are considered satisfied if objects are placed within a certain threshold of their target location.

# E    Additional Results

This section presents additional results and analyses of HA. We first analyze the sensitivity of task performance to several hyperparameters used in HA from creating the graph: the number of clusters, number of buffer size, and the number of slots used in slot attention. We next tested whether giving our model-based baselines more computation time compared to HA would improve their performance to be comparable to HA's.

## E.1    Analysis of key hyperparameters

In this section, we analyze the sensitivity of our method to various hyperparameters, evaluated in the robogym environment with four objects in the complete goal specification. As Fig. 7 shows, performance depends on the number of initialized clusters and the number of batches from the training set used to construct the graph. With too few clusters, the clusters are too coarse-grained to differentiate objects in significantly different positions, while with too many the performance deteriorates as the data is needlessly split into duplicate clusters. Performance improves with more data, as the graph has better coverage. Although our model performs worse when there are insufficient slots to represent all objects present in the environment plus one empty slot, performance is barely impacted by having double the number of necessary slots. Our method can thus still work in environments with an unknown but upper-bounded number of objects.

## E.2    More challenging evaluation settings

We analyzed HA in more challenging settings that reflect the noisy nature of real-world robotics. As Fig. 8 (left) shows, HA is robust to the addition of Gaussian noise to the action at every time step, up until the noise variance is comparable to the maximum distance for successful picking and goal placements. The performance also remains high given significantly fewer steps (Fig. 8, right).

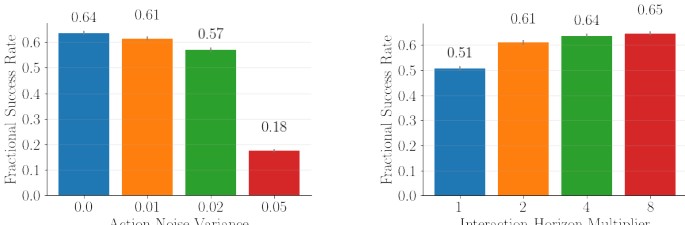

Figure 8: The performance of HA on *robogym-rearrange* as we vary the amount of noise added to the actions, and the interaction horizon (as a multiple of the minimum steps needed to complete the task).

