# OpenReview forum: "Hierarchical Abstraction for Combinatorial Generalization in Object Rearrangement"
_NeurIPS.cc/2022/Workshop/SVRHM — SVRHM Poster_

### Official Review · Reviewer_Zjjk · 2022-10-08
**An approach to solving blockworld problems with a transformer architecture that uses a prior on entities.  The approach seems solid but lacks some comparison to previous work and a connection to the topic of SVRHM**

**Rating:** 6
**Confidence:** 3

**Review:**

The authors present a functional and tested method for automated processing of video content to build a representation that separately tracks the entities in the video as well as their transitions.  Rather than a conventional approach of having a monolithic architecture that processes video content for prediction, this approach uses a prior that entities and transitions are separate representations. The approach is validated against existing approaches to processing these blocks-world style problems and compares favorably.

It seems likely that other approaches have used similar if not identical approaches to disentangling identity and object representations.  The approach has roots in the class SHRDLU symbolic approach to AI which essentially solved this class of problems for entities that are discrete.  Reference to this work would seem appropriate.  There are other deep learning approaches to this class of problems already, and it would be helpful to have a better sense of how the approach is unique relative to other entity based models such as OP3 (Veerapeneni et al.).

It is also unclear why the existing baseline comparisons here are performing so poorly relative to the new approach, when they have already been shown to solve a similar class of problems in their original publications.

The paper would benefit from a clearer description of several aspects including:

Clearer description of whether the objects are being parsed directly from the pixelated input or if object segmentation is pre-specified.  Likewise, is the set of transitions learned?

What exactly is the output and how is it compared against ground truth to compute an error for training?

Why are existing approaches doing so badly at this specific task which seems similar to tasks they are already demonstrated on?  Is it because there are only two frames rather than a video?  I wonder if the enormous difference in performance between this approach and e.g. MPC is due to a genuine functional capability or a lack of calibration of MPC.  I.e. would a slight tweak of MPC on this particular objective function allow it to perform much better?

Figure 1 is difficult to parse especially panel d.  In panel e, shouldn’t the goal state be more than 1 step different than the starting state?  What do the bicolor circles represent?   Smallest fonts should be larger since even laser printers struggle to print these small letters.

In the evaluation, how many different actions were required between the starting and goal state?

Another issue that is important and quite relevant to this particular workshop is that there is no effort made to connect the approach to human cognition, or neural representations.  For example the constraint that identities are stable over time is akin to the principle of object-permanence, which we take as a key developmental milestone in the development of human children.

---

### Official Review · Reviewer_F7Pf · 2022-10-10
**A new method to solve object rearrangement tasks**

**Rating:** 7
**Confidence:** 1

**Review:**

The paper proposes a new method to solve object rearrangement tasks. With these methods, agents can generalize the rearrangement strategy that can be learned from sensorimotor interface. The results showed a higher performance than other RL methods for these tasks.
I think the performance results are exciting and the paper was understandable clearly. However, human performance was not mentioned (rather than task succession, task solving speed can be mentioned). In the discussion part, I would like to see the explanations why focusing on how well inferred entities can reused for solving tasks performed better than other RL models more and applicability of this model for more embodied (like robotic) area.

---

### Official Review · Reviewer_cgVX · 2022-10-14
**Seems like a valuable contribution but not my area of expertise**

**Rating:** 7
**Confidence:** 1

**Review:**

The topic of this work is not my area of expertise so my understanding is limited. The HA method appears to achieve significant gains in combinatorial generalization over the baseline methods.

Quality: Not confident enough to comment
Clarity: The paper is written clearly aside from:
Typo in figure 4: “ataset”
Figure 1 fonts are too small
Originality: Not confident enough to comment
Significance: Not confident enough to comment

---

### Official Review · Reviewer_aCrm · 2022-10-14
**The problem of object rearrangement is extremely important for understanding the true concepts of objects and attributes that are crucial for generalization. The approach is interesting but the paper could have been illustrated in a better way to clear some points mentioned below. The paper fits the scope of the workshop.**

**Rating:** 6
**Confidence:** 2

**Review:**

In this work, the authors propose a hierarchical abreaction (HA) approach to address the problem of object rearrangement. The aim of this work is to enable learning-based systems to reuse learnt data by generalizing over a combinatorially  large space of object configurations. HA consists of a hierarchical modeling component and a control component. The modeling component abstracts the training data into a factorized graph at two levels. In the first level, the nodes in the graph are created to represent object entities with an action independent identity and an action dependent state. In the second level the nodes are connected to form a state transition graph with the edges representing the single action that transforms a node’s state to another. The control component can then be used during inference. In this approach, the object configurations are first abstracted to match the nodes of the state transition graph. Then the appropriate action is taken to transition from the current image observation to the goal observation. The implemented technique was evaluated four environments with object rearrangement tasks and compared to existing methods. The authors claim that their approach shows about a 5-10x improvement in the combinatorial generalization accuracy.

Pros:
    1. The proposed approach provides a key intuition about data abstraction to enable knowledge reuse. This abstraction is learnt and does not need human intervention, resulting in a methodology that can be extended to other tasks.
    2. Hierarchical abstraction is not only better at generalization, but also computationally less expensive to run compared to existing offline reinforcement learning techniques.

Cons:
    1. The authors can improve the paper by providing some intuition or answers to the following questions.
        a. The criteria abstracting visual features into sets of entities enforces that the transition should be sparse, specifically only one entity can change in the transition.  However, this seems like a highly restrictive environment. How would HA (or a small extension of HA) perform when there are more entities changing in the transition?
        b. Since the modeling approach only learns on the one entity that is changing, what is the need to/advantage of considering a dataset with multiple objects?
        c. The proposed control component involves a clustering state transitions, as well as a permuting over all the entities to align them. The computational cost of both of these steps depends on the number of entities in question. Would the algorithm lose out its advantage of having a lower computational cost over existing techniques as the number of entities increase?
        d. Do concepts from Causality to disentangle objects from attributes be incorporated in any way into this approach?